# Empirical Study on the Influence of Urban Environmental Industrial Structure Optimization on Ecological Landscape Greening Construction

**DOI:** 10.3390/ijerph192416842

**Published:** 2022-12-15

**Authors:** Lili Yang, Ning Ma

**Affiliations:** 1School of Art and Design, Shandong Women’s College, Jinan 250300, China; 2School of Economics and Management, Wuhan University, Wuhan 430072, China

**Keywords:** urban environment, optimization of industrial structure, ecological landscape, green construction

## Abstract

With the rapid development of the economy in China, the ecological environment problem of the city has become an important factor that restricts the development of our economy and society. People gradually realize that, while rapidly generating wealth, they have been shrouded by the shadow of environmental pollution for a long time, which makes people feel more and more frightened and thoughtful. Industry is the carrier of economic activities, so we must pay attention to the relationship between industry and the natural environment. In this case, people pay more and more attention to the study of ecological construction and bring it into the optimization of environmental industrial structure. During this period, to correctly handle the relationship between industrial structure and ecological construction, to realize the overall transformation, development and cultivation of industrial structure, is the inevitable choice to promote the healthy and sustainable development of enterprises. From the perspective of industrial structure adjustment and the green space system, this paper makes theoretical assumptions about the impact of environmental industrial structure adjustment on urban ecological green space construction. Then, through the panel data of 260 cities from 2008 to 2018, the impact of China’s industrial structure adjustment on the scale of urban ecological green space was empirically analyzed. On this basis, this paper puts forward some policy recommendations for the development of urban ecological green space in our country.

## 1. Introduction

As a result of the country’s reform and opening up over the past thirty years, China’s economy has developed rapidly and achieved remarkable results, and by 2010 it had leaped to the ranks of the second largest economy in the world. However, with the rapid development of China’s economy, the degradation of the ecosystem, the tightening of resource constraints and serious environmental pollution have led to the increasingly obvious contradiction between human beings and the ecological environment [1]. Ecological environment is a worldwide problem, which is related to the development and destiny of human beings, and the traditional industrial development model, which takes economic development as the ultimate goal, is facing serious challenges [2]. 2012, the 18th National Congress included the construction of ecological civilization into the national “five-in-one” strategic plan, and in 2015, the Fifth Plenary Session of the Communist Party of China included the strengthening of ecological civilization into the five-year plan for the first time. Ecological construction is a new form of civilization beyond industrial civilization, which is based on ecological civilization and cannot be built without the support and guidance of ecological culture construction [3].

The lack of understanding of ecological culture has led to serious pollution and destruction of our natural environment, and only a deep understanding of the concept of ecological culture can enhance the awareness of ecological environment protection [4]. In the development of ecological economy, the limitations of the traditional theory and practice of industrial structure optimization have gradually emerged. In the 12th Five-Year Plan, the adjustment of economic structure is one of the ten major tasks in the 13th Five-Year Plan. The optimization of industrial structure is directly related to the quality and efficiency of China’s future economic and social development [5]. In this context, ecological construction has put forward new requirements for the adjustment of industrial structure, namely, to promote the optimization and upgrading of industrial structure on the basis of the harmonious development of humanity and nature, and to form an ecological industrial system, but also to make it an important source of economic development [6]. In the process of achieving economic development, the relationship between the industrial structure and the construction of ecological civilization should be properly handled to achieve a new environmental industrial structure compatible with coordinated and sustainable development.

In summary, with the continuous development of economic development, environmental pollution and ecological environment deterioration are leading to the increasingly prominent problem of resource and environmental constraints on economic development, and the development direction of environmental industrial structure must be properly grasped to ensure sustainable economic development [7]. In this paper, based on 11 years of historical data of 260 prefecture-level cities, we conducted an in-depth study on the optimization of urban environmental industrial structure, analyzed its impact on urban ecological landscape greening construction, and put forward corresponding policy recommendations based on the empirical results.

## 2. Research Hypothesis

According to Wang and Chen (2003), industrial structure is an important link between human socio-economic and ecological environment. From the perspective of production, industrial structure is a “resource allocation device”, and from the perspective of environmental protection, it is a “control body” of environmental resources, which is the type and quantity of pollutant industries [8]. Therefore, in the micro-environmental management, especially for the serious environmental problems faced by big cities with severe pollution, people turn their attention to the adjustment of industrial structure. Analysis of the complex relationship between two interacting subsystems, the industrial environment and the urban environment, is a prerequisite for industrial restructuring in China [9,10,11]. According to Zhang, Cao, and Dong (2021), urbanization is a major source of driving force for economic development, and its degree of development is an important indicator of the overall development of a region [12]. While urbanization is driving economic development, advancing social progress, and improving people’s livelihood, it has also brought about increasingly prominent environmental pollution problems; the development of high-consumption and high-pollution industries has brought about an increase in environmental load, the roughness of urbanization has made it more difficult to protect the environment, and achieving the coordinated development of urbanization, industrial restructuring, and ecological environment is an urgent problem [13,14,15]. Currently, China is in an important stage of industrial transformation, and the country is gradually reducing the proportion of industries and developing high-yield, low-pollution industries, such as service industries and high-tech industries, to reduce environmental pressure, and the transformation and upgrading of industrial structure has become an important factor in promoting urbanization and environmental protection [16,17,18]. Zhang (2009) proposed that the optimization of industrial structure oriented to ecological cities is the key to economic structure optimization only in terms of the mechanism of the influence of industrial structure change on urban economic growth, which is to bring resource factors together to high-productivity industries [19]. This criterion has been overused in the practice of industrialization in many countries, especially in developing countries, which has led to three problems in our ecological development: first, the imbalance of our economic structure due to the implementation of an unbalanced industrial policy [20,21]. The coordination of the proportion of the industrial structure is an ecological problem, and only the coordination of the proportion between different resource elements in different sectors and links can make different resources compensated in different value forms and material forms, so as to achieve the maximum use of resources and prevent structural waste of resources [22]. Second, the ecological benefits of advantageous industries are not guaranteed. Heavy industry, high pollution, high energy, resource development and other industries are the core industries of some cities, which develop abnormally, leading to the concentration of resources and the intensification of damage to the environment [23,24,25]. Third, the simple emphasis on reducing the proportion of secondary industries and accelerating the development of tertiary industries is only to increase the ecological footprint of the city, which is trying to achieve the best environment but not sustainable development completely free from the constraints of resources and environment [26,27]. After a short period of rapid development, due to changes in the industrial structure, there are inevitable resource and environmental constraints that must be addressed for another economic shift, which will come at a great cost, even irreversible decline [28]. In their study, Liu and Liu (2021) argue that to evaluate the role of “civilized cities” on industrial structure upgrading, two aspects must be addressed: first, the intrinsic factors, the selection of civilized cities and industrial structure upgrading have an inverse intrinsic link, while the selection of civilized cities will affect the direction of investment, production scale and technological innovation, thus having a certain influence on the development of the industry; the upgrading of industrial structure will directly affect the decision making of an enterprise and the overall development of a city, thus determining the civilization of a city; secondly, the mechanism of evaluating the role of civilized cities on the upgrading of industrial structure is still unclear; the selection of civilized cities is an important measure to promote the construction of urban civilization in China, which will be evaluated through a series of strict indicators and a set of perfect evaluation processes to enhance the civilization degree of the whole city [29,30,31,32,33,34]. On the surface, the evaluation of a civilized city will only affect the cultural level of a city and not the development of industries. In fact, the evaluation of civilized cities must have a more complex transmission mechanism for the optimization of industrial structure [35,36,37,38,39]. The selection of civilized cities can both improve the living environment of residents and promote sustainable economic development, and will also generate potential demand for technological innovation and green development in cities, while it will have an important impact on the transformation and upgrading of China’s environmental industrial structure [40,41,42,43]. Based on the above analysis, the following hypotheses are proposed in this paper.

**Hypothesis** **1.**
*The optimization of urban environmental industrial structure has a positive impact on the construction of ecological landscape greening.*


## 3. Research Design

### 3.1. Data Sources

In this study, balanced panel data based on data from China’s Urban Statistical Yearbook 2008–2018, China’s Urban Construction Statistical Yearbook 2008–2018 and statistical yearbooks of all provinces in China were used. The total number of samples was 2750, covering the statistical data of 260 cities in China for 11 years from 2008 to 2018. In this paper, the average interpolation method is used to deal with the missing report in some areas, and the data of Hong Kong, Macao and Taiwan areas are excluded according to the consistency principle of statistical indicators. Hainan, Xinjiang and Tibet also had a large number of missing data and were also excluded.

### 3.2. Variable Selection

#### 3.2.1. Explained Variables

Based on the greening area of construction land (LS), the urban ecological greening scale in China has been analyzed in this paper. The green coverage rate (GS) of the built-up area was taken as the main alternative index, and the green area was compared and analyzed.

#### 3.2.2. Core Explanatory Variables

The adjustment of environmental industrial structure is a dynamic process, which is divided into two levels: “advanced” and “rationalization”.

Among them, the advanced environmental industrial structure (*INDS*) uses the ratio of tertiary industry GDP to secondary industry GDP as a measure. Currently, the economic structure of our country is undergoing a transition to a service-oriented one, which is a distinctive feature of the upgrading of the industrial structure, that is, the growth rate of the value added of the industry is accelerated compared to the output of the secondary industry [44,45,46,47,48,49]. Through this index, we can clearly see the transition of the economic structure of our country from the low end to the high level, which also clearly reflects the tendency of the industrial structure to be “service-oriented”, thus becoming a better indicator of the advanced industrial structure. If the *INDS* index tends to increase in this study, it indicates that our economy is moving toward a service-oriented development and a higher industrial structure [50,51,52,53].

The rationalization of environmental industrial structure (*TL*) uses the Thiel index (*TL*) to measure the rationality of industrial structure and to refine it. The connotation of industrial structure rationalization refers to the structural transformation ability between different industries, the degree of efficient use of resources, and the coordination between the structures of each factor input and output [54,55,56,57,58]. The industrial structure rational output value proposed in this paper is a deviation index, which reflects the negative correlation of industrial structure rationalization, and the larger its value is, the lower the degree of industrial structure rationalization is. It is calculated by the formula:$$TL_{it}=\sum_{k=1}^{3}\left(\frac{Y_{itk}}{Y_{it}}\right)\ln\left(\frac{Y_{itk}}{L_{itk}}/\frac{Y_{it}}{L_{it}}\right)$$

In this paper, *TL* represent the reasonable deviation of the environmental industry structure, *i* represents the region, *t* represents the year, Y represents the output value, *L* represents the employees, and *k* represents the industry; this paper takes the primary, secondary and tertiary industries as the research objects, so 1–3 is used to find *k*. $Y_{i}/Y$ shows the composition of the yield. Y/L shows productivity. When the economy is in equilibrium, *TL* is 0. On the contrary, *TL* is not equal to 0 in the case of industrial structure change. It can be seen that the higher the *TL* value, the worse the rationality of the industrial structure.

#### 3.2.3. Control Variables

As shown in Table 1, the model takes construction land (*BS*), total population at the end of the year (*Popu*), provincial financial funds (*FIAN*), per capita GDP (*Per_gdp*), and garden and green space investment (*Park_invest*) as variables.

### 3.3. Regression Model

The multiple linear regression model was established, and the regression analysis was used to test the relevant variables in this study:$$\begin{array}{l} LS_{it}=\beta_{0}+\beta_{1}*INDS_{it}+\beta_{2}*BS_{it}+\beta_{3}*Popu_{it}+\beta_{4}*Fian_{it}+\\ \beta_{5}*Per\_gdp_{it}+\beta_{6}*Park\_invest_{it}+\varepsilon_{it} \end{array}$$
$$\begin{array}{l} LS_{it}=\gamma_{0}+\gamma_{1}*TL_{it}+\gamma_{2}*BS_{it}+\gamma_{3}*Popu_{it}+\gamma_{4}*Fian_{it}+\\ \gamma_{5}*Per\_gdp_{it}+\gamma_{6}*Park\_invest_{it}+\varepsilon_{it} \end{array}$$

The *LS* explained variable was used to measure the ecological green area of the city. $\beta_{0}$ and $\gamma_{0}$ are the segmentation term of the model. $INDS_{it}$ is an important explanatory variable which has a high research value. $TL_{it}$ is the core explanatory variable used to measure the rationality bias of the corporate structure. The direction and degree of action of *INDS* on *LS* was measured by $\beta_{1}$. $\gamma_{1}$ can measure the direction and degree of influence of *TL* on *LS*. All the other variables are control variables; $\beta_{2}-\beta_{6}$ and $\gamma_{2}-\gamma_{6}$ are the corresponding coefficients of each control variable. $\varepsilon_{it}$ is a random jamming project. The subscript *i* is the interval, the cities across the country, and the *t* is the time, the different periods studied.

In order to ensure the correctness and unbiased nature of the empirical analysis, it is necessary to investigate the inherent problems in the model. In addition to social and economic reasons, the difference of ecological green area between different prefectures is also related to the natural geographical environment. For example, Zhongwei city in Ningxia Province, as it is an arid desert area, has a certain green level gap with Nanjing city in Jiangsu Province. Our vast territory and big cities produce differences due to their natural conditions, and there are also many manifestations in the sampling data of this paper. The error caused by the physical geographical conditions cannot be controlled by the parameter setting in this paper. If the mixed regression model is used, the accuracy of the model may be reduced due to the inherent problems caused by the omission of variables, so that the results of coefficient estimation are not credible.

In order to improve the accuracy of empirical analysis, this paper adopts a two-way fixed effect model of time individual and eliminates the influence of natural geographical factors on the internal environment.

## 4. Empirical Analysis

### 4.1. Descriptive Statistical Analysis

As shown in Table 2, the authors compiled data from a sample of 2750 and used SPSS software to statistically analyze the data to produce descriptive analysis of the indicators.

The green area of the built-up area is 513,166,507 square meters at the most and 220,011 square meters at the least, which is far from it. Likewise, there is a great difference in the greening coverage rate of urban construction land. It shows that the ecological greening level of different cities varies greatly from the whole country.

As can be seen from the data in Table 2, the standard deviations of the individual variables are quite large. It shows that the dispersion of some original data is very strong. The average value of the core explanatory variable, environmental industrial structure upgrading (*INDS*), was 0.988. It shows that the total industrial production value of Chinese cities is lower than the GDP of the primary industry on average. However, the maximum value of this variable is 5.072, indicating that in all prefecture-level cities, the tertiary industry is the main industry driving regional GDP, but its development speed is relatively slow due to regional and other constraints.

To exclude the problem of multicollinearity among the variables, the authors performed a correlation analysis. As shown in Table 3, it is clear from the data in the table that none of the correlation coefficients among the remaining control variables is greater than 0.7, and therefore, there is no multicollinearity problem among these variables.

Firstly, the relationship between the core explanatory variables and the dependent variable was investigated: the correlation between industrial structure advanced (*INDS*) and green area of built-up areas (LS) was 0.165 and significant at the 1% level, indicating that there is a significant positive correlation between the two, and industrial structure optimization can promote the green area. The correlation between the deviation of environmental industrial structure rationalization (*TL*) and the green area of built-up areas was −0.438 and had a significant f correlation at the 1% level, indicating that the larger the deviation of industrial structure rationalization, the greater the inhibitory effect on the green area. Secondly, the effects of construction land, population at the end of the built-up area, provincial financial resources, GDP per capita, and fixed asset investment in landscaping on the green area of the built-up area were examined, and the results show that all these factors had a significant positive effect on the green area of landscaping, and hypothesis 1 was initially verified.

### 4.2. Basic Regression Analysis

To further test the hypothesis of this paper, the authors conducted an econometric regression analysis based on the data information, and Table 4 shows the results of the regression. On this basis, the method of regulating urban green space area by dummy variable is put forward. In order not to make the table seem too lengthy, the coefficients of the individual dummy variables are not reported and are instead expressed as “controls”, where the effects of individual and time are controlled in the regression results. In addition, to eliminate the influence of heteroscedasticity on the model in this paper, the clustering robust standard error of city level is introduced into the model to ensure the reliability of the model.

The regression analysis was concluded as follows: the regression analysis in result (1) showed an *INDS* coefficient of 0.446 with significance at the 1% level even without the effect of other factors. On this basis, each control variable was included in the regression analysis separately for the purpose of controlling for other factors. The results show that (2) to (6) each *INDS* variable has a significant test, and the coefficient of influence of the *TL* factor in result (7) is −0.265 and is statistically significant at the 1% level, and all the predicted values are above 0.4, indicating that the control variables selected for this study are relatively reasonable.

The *INDS* coefficient in result (6) is 0.507, which is significant at the 1% level, that is, if other variables are controlled, then the increase in the proportion of secondary and tertiary industries significantly contributes to the increase in the area of landscape greenery. The results show that there is a significant positive correlation between the scale of urban landscape greening construction and the advanced industrial structure during the study period, and this finding is consistent with hypothesis 1 of this paper.

The *TL* coefficient in result (7) is −0.265, which is significant at the 1% level, i.e., when other factors are controlled, each unit increase in the index of reasonable deviation of environmental industrial structure leads to a 26.5% decrease in the green area within the construction area. The results show that there is a significant negative correlation between the deviation of the rationality of industrial structure and the scale of urban ecological green space among regions.

The results (6) and (7) validate Hypothesis 1.

In terms of control variables, built-up areas, year-end population in municipal districts, GDP per capita in municipal districts, and fixed asset investment in landscaping were all subjected to correlation analysis, indicating that all these indicators have a significant positive correlation with the ecological green space scale of the city. Taking the area variable term of built-up area in the municipal district as an example, its influence factor is about 0.041, which indicates that when the area of built-up area in the municipal district increases by 1 unit, then the green area of green space can be increased by about 4.1% on average; thus, the elasticity magnitude among the indicators can be used as an indicator under the same control of other factors. In addition, the population ratio at the end of the urban area also showed an obvious positive trend, which is the growth of the number of urban population and the demand for urban ecological greenery, which is also the pursuit of green living space by urban residents [59,60,61,62].

However, the inter-provincial fiscal allocation index was not tested for significance, which suggests that there is not enough evidence to confirm what effect provincial fiscal allocations have on the scale of urban ecological green spaces at this study stage. It has been argued that the ratio of national income to labor force in the tertiary sector gradually increases as the industrial structure deepens, while the ratio of national income to labor force in the primary sector gradually decreases, and it can be seen that the labor capacity of the tertiary sector can be improved through the adjustment of the environmental industrial structure [63,64,65,66,67]. During the past decade, China’s economy is in a transitional stage from industrialization to diversification, and with the development of the economy, the environmental industrial structure is readjusted and the labor force is transferred from secondary to tertiary industries, and the analysis of the impact of urban population on the demand for ecological green space yields the demand for urban ecological green space from the concentration of tertiary industrial population, which increases the green area of built-up areas [68,69,70,71,72].

The reliability of the above findings is also to be tested by introducing alternative measures of the explainable variables. The following are robustness tests.

### 4.3. Robustness Test

To test the robustness of the above findings, we used the green coverage (GS) of built-up areas as an alternative metric. Table 5 shows the results of the robustness analysis.

The results show that the *INDS* coefficient is 0.503 and the *TL* coefficient is −0.258, both of which are significant at the 1% level, and the goodness of fit is 0.916 and 0.937, respectively. This indicates that the adjustment of environmental industrial structure has a significant positive impact on the construction of urban ecological landscape greening. Among the other control variables, the coefficients of all factors are positive and the degree of influence is relatively stable, so the conclusions of this paper are robust and reliable [73,74,75,76,77].

### 4.4. Heterogeneity Analysis

Due to the vast territory of China, it can be seen from previous studies that there is a certain difference in the level of optimization of the environmental industrial structure among regions, so it can be inferred that there are differences in the impact of the ecological building layout on different regions. In general, in economically developed regions and regions with more relaxed economic development, the transition from “primary industry” to “tertiary industry” and “industrial economy” to “ecological economy” has been a very important factor. In the transition period of “ecological economy”, ecological greening will be expanded accordingly to further enhance the image of the city, improve the living environment of the city and attract talents; while in less developed areas, especially under the constraints of geographical environment, the secondary industry is still the main pillar of its economic development. Due to the high price of land, the problems of “cutting mountains to make cities” and “filling land to make cities” have occurred in order to expand urban space, thus inhibiting the expansion of ecological construction in cities [78,79,80,81]. Therefore, for this type of region, the pull of environmental industrial restructuring on the scale of urban green space has not been realized yet.

From the analysis of the above information, it can be seen that there are different regions in the east, middle and west of China; thus, the information of each region of the country is divided into three regions, east, middle and west, and the econometric regressions are conducted separately to explore the influence of environmental industrial structure optimization movement on urban landscape greening construction. The detailed results are shown in Table 6.

The above table shows the regression analysis for the three regions of East, Central and West. Firstly, the factors of advanced environmental industrial structure were analyzed, from the regression results of the eastern region, the *INDS* index was 0.424 and was tested at a significance level of 10% or more, indicating that the advanced environmental industrial structure plays a greater role in promoting the growth of urban ecological green space. The analysis of the advanced industrial structure in the central region shows a significant positive effect at the 1% level, with a coefficient of 0.892, indicating that the advanced environmental industrial structure in the central region has a greater driving effect on the growth of urban green space. The results of the analysis of the central region’s industrial structure plan show a positive influence at the 1% level with a coefficient of 0.892, indicating that the advanced environmental industrial structure in the central region has a greater driving effect on the growth of urban green space; the *INDS* index of the western region is 0.038, which does not have a statistically significant level, indicating that the advanced degree of environmental industrial structure in the region does not have a significant influence on the area of regional ecological green space. Secondly, the analysis of industrial structure rationalization, the regression analysis results of east and west do not have statistical significance at the level of statistical significance, and the index of ecological green area in the central region is −0.462 and passed the test at the 5% significance level, which indicates that the change of the rational degree of environmental industrial structure in the central region has a significant negative impact on the construction of landscape greenery, and for each unit increase in the deviation of industrial structure rationalization, the respective value of landscape greening construction decreases by 46.2%.

The above analysis shows that there is a difference in the impact of environmental industrial structure optimization on the scale of urban landscape green space in the three regions of eastern, central and western China, which is not obvious in the regions with lower and higher economic levels.

## 5. Conclusions and Suggestions

### 5.1. Conclusions

The empirical analysis shows that the adjustment of environmental industrial structure has a significant positive effect on the scale of urban ecological space during 2008–2018. In these indicators, with the high-grade environmental industrial structure, the urban ecological green area also increases. With the increase in the reasonable deviation degree of the environmental industrial structure, the ecological green area of the city also decreases. In addition, urban population, per capita output value and urban green investment are positively correlated with the scale of urban ecological green space. Through the analysis of regional heterogeneity, it has been found that the influence of industrial structure change on urban green space area shows heterogeneity. Its main characteristics are: in the eastern region, with the continuous improvement of the level of economic development, the urban green space will be significantly improved; in the western underdeveloped areas, there is not so significant an effect.

#### 5.1.1. Overall Research Results

From 2008 to 2018, the optimization of urban environmental industrial structure has reflected a significant role in promoting the construction of urban ecological landscape greenery. In this paper, urban environmental industrial structure is mainly measured by two indicators, advanced industrial structure and rationalization, and based on the results of empirical analysis, it can be seen that the coefficient between advanced industrial structure (*INDS*) and ecological landscape greening area (LS) is 0.165, with a significant positive correlation at the 1% level, indicating that industrial structure optimization can promote the ecological landscape greening area and environmental industrial structure. The coefficient between the rationalization of environmental industry structure (*TL*) and ecological landscape greening area (LS) is −0.438, with a significant negative correlation at the 1% level, indicating that the greater the deviation of industrial structure rationalization, the greater the inhibitory effect on landscape greening construction.

#### 5.1.2. Findings from Subregional Studies

In the three regions of East, Central and West, the impact of the adjustment of environmental industrial structure on urban ecological space is different. Firstly, the analysis of advanced environmental industrial structure and according to the results of regional regression analysis, it can be seen that the *INDS* coefficient from the eastern region is 0.424, with significance at the 10% level, indicating that the advanced environmental industrial structure in this region plays a greater role in promoting the growth of urban ecological green space; the *INDS* coefficient in the central region is 0.892, with significance at the 1% level. The advanced environmental industrial structure plays a more significant role in promoting the growth of urban ecological green space; the *INDS* index of the western region is 0.038, which does not have significance at the statistical level, indicating that the degree of advanced environmental industrial structure in this region does not have a significant impact on the area of regional ecological green space. Secondly, the analysis of industrial structure rationalization, the regression analysis results of the east and west do not have statistical significance at the level, and the index of ecological green area in the central region is −0.462 and passes the test at the 5% significance level, which indicates that the change of the rational degree of environmental industrial structure in the central region has a significant negative impact on the construction of landscape greenery, and the enhancement of rationalization deviation will significantly inhibit the construction of landscape greenery. The increase in rationalization deviation will significantly inhibit the improvement of landscape greening construction. The overall analysis shows that the optimization of urban environmental industry structure has less influence on the landscape greening construction in the western region with lower economic level and the eastern region with a developed economy, and the influence on the landscape greening construction in the central region is relatively more obvious, and the western, central and eastern regions reflect obvious regional differences.

### 5.2. Suggestion 

#### 5.2.1. Government Policy Support

The government should vigorously support state-owned and listed companies with large-scale and strong technical strength to actively participate in the implementation of urban greening projects. During the construction process, the construction unit should be urged to improve production equipment, save energy, improve the technical quality of greening projects, and enhance the positive role of greening projects in terms of operational efficiency, reducing greening maintenance costs, and increasing the value-added of green space services. In addition, foreign capital can be appropriately encouraged to invest in urban ecological greening construction to accelerate the mutual transformation of various elements, thus making China’s green industrial structure more in line with the international environment; at the same time, units engaged in green building projects should be encouraged to actively absorb advanced foreign design ideas and production processes in order to improve the technical level and service effectiveness of green buildings [82,83,84,85,86,87].

#### 5.2.2. Deepening Structural Adjustment

The relationship between industries should be properly handled so that each industry can develop in a coordinated manner and form a leading industry with high economic and ecological benefits. Firstly, the restructuring of the three industries should gradually shift from the secondary industry to the tertiary industry according to the city’s resources, talents, technology and economic development level, while maintaining a reasonable structure. Secondly, the relationship between the development of high-tech and traditional industries should be correctly handled, and new industries such as electronic information, bio-engineering and new materials should be taken as new economic growth points; high-tech, advanced and practical technologies should be used to transform traditional industries and enhance the development momentum of enterprises. Again, the relationship between technology-intensive industries and capital-intensive industries should be correctly handled, and the development of capital- and technology-intensive industries should be emphasized to improve the ecological benefits of the city. Finally, the development of ecological cities cannot be completely separated from the sustainable and healthy development of the real economy; it is necessary to actively guide the development of the virtual economy on the basis of industrial linkages and actively use it to promote the development of the economy, improve the overall economic efficiency of the economy, and reduce the consumption of materials and energy [88,89,90,91,92,93].

#### 5.2.3. Refine the Industrial Division of Labor

According to the basic theory of circular economy, a symbiotic network of industries is constructed so that the two-way loss of industrial development to the natural environment can be minimized. To improve the effectiveness of urban ecosystems, the traditional one-way linear coupling between industrial systems, resource systems and environmental systems must be radically changed to form a circular compensatory cycle coupling relationship. The production, supply and disposal systems of industrial systems must be designed and operated in a way that restores and maintains the quality of the environment. Specifically, at the macro level, the division of labor among industries should be further deepened and close industry linkages should be formed under the guidance of urban industrial policies and market mechanisms; at the meso level, major industries should be integrated into an organized circulation system. At the micro level, clean production should be promoted within enterprises to achieve material and energy cycles in the industry [94,95,96,97].

#### 5.2.4. Development of Ecological Industries

The aim of this study was to find a way to improve the quality of the environmental industry and promote the adjustment of industrial structure and ecological transformation. Industrial ecology is a kind of management of traditional industries based on ecological economy, using ecological, economic and system engineering to achieve the maximum socio-economic benefits, to achieve the maximum use of resources as well as to reduce the damage to the ecological environment and to realize the multi-level use of resources. Technically speaking, there are no pure industries, only clean products and services. Therefore, industrial ecologization must be based on upgrading the internal structure and ecological environment of the industry, taking scientific and technological progress as the premise, and further realizing the ecological transformation of the industry from micro to macro and from technology to management of the whole process [98,99,100,101].

## Figures and Tables

**Table 1 ijerph-19-16842-t001:** Variable definition table.

Symbol	Type	Meaning
LS	Explained variables	Log form of green area in built-up area
GS	Alternative indicators of robustness	Logarithmic form of green space area of construction land
INDS	Core explanatory variable	The industrial structure is advanced
TL		Rationalization of industrial structure
BS		Logarithmic form of construction land
Popu	Control variables	The total population of the municipal district at the end of the year in logarithmic form
Fian		Logarithmic form of provincial financial funds
Per_gdp		Per capita GDP in log form
Park_invest		Logarithmic form of garden green space investment

**Table 2 ijerph-19-16842-t002:** Descriptive statistical results.

The Variable Name	The Number of	The Average	The Standard Deviation	The Minimum Value	The Median	The Maximum
LS	2750	17.138	1.040	12.323	17.087	20.078
GS	2750	17.261	1.10	12.479	17.185	20.295
INDS	2750	0.988	0.596	0.105	0.849	5.072
TL	2750	0.927	0.224	0	0.880	3.123
BS	2750	13.44	1.066	8.093	13.104	17.858
Popu	2750	13.830	0.797	11.915	13.749	17.025
Fian	2750	7.400	1.975	1.110	7.403	14.458
Per_gdp	2750	10.70	0.760	2.813	10.531	13.719
Park_Invest	2750	18.544	1.789	11.929	18.570	23.467

**Table 3 ijerph-19-16842-t003:** Correlation coefficient matrix.

	LS	INDS	TL	BS	Popu	Fian	Per_gdp	Park_Invest
LS	1							
INDS	0.165 ***	1						
TL	−0.438 ***	0.084 ***	1					
BS	0.419 ***	0.020	−0.263 ***	1				
Popu	0.689 **	0.096 **	−0.121 **	0.414 ***	1			
Fian	0.216 ***	0.044 *	−0.097 ***	0.167 ***	0.227 ***	1		
Per_gdp	0.607 **	−0.156 **	−0.595 **	0.251 ***	0.267 ***	0.152 ***	1	
Park_Invest	0.494 ***	0.058 ***	−0.238 ***	0.265 ***	0.468 ***	0.158 ***	0.414 ***	1

Note: *, ** and *** denote significance at the 10%, 5% and 1% levels, respectively.

**Table 4 ijerph-19-16842-t004:** Results of the bidirectional fixed effect model.

Explained Variable: LS							
	(1)	(2)	(3)	(4)	(5)	(6)	(7)
INDS	0.446 ***	0.451 ***	0.467 ***	0.469 ***	0.500 ***	0.507 ***	
	(3.94)	(3.905)	(4.050)	(4.074)	(4.352)	(4.458)	
TL							−0.265 ***
							(2.677)
BS		0.043 ***	0.043 ***	0.042 ***	0.040 ***	0.041 ***	0.036 ***
		(4.050)	(3.974)	(3.924)	(3.187)	(3.897)	(3.014)
Popu			0.244 ***	0.246 ***	0.293 ***	0.284 ***	0.185 ***
			(3.803)	(3.93)	(4.313)	(4.327)	(2.548)
Fian				0.019	0.019	0.018	0.013
				(1.297)	(1.284)	(1.140)	(0.36)
Per_gdp					0.180 **	0.175 **	0.073
					(2.232)	(2.318)	(1.438)
Park_Invest						0.044 ***	0.040 ***
						(5.093)	−4.649
_cons	16.289 ***	15.858 ***	12.648 ***	12.566 ***	10.263 ***	9.844 ***	13.085 ***
	(150.096)	(108.823)	(14.366)	(14.195)	(7.318)	(7.35)	(11.656)
Individual	control	control	control	control	control	control	control
Year	control	control	control	control	control	control	control
R—Square	0.69	0.496	0.503	0.503	0.513	0.521	0.448
	2750	2750	2750	2750	2750	2750	2750

Note: (1) t value in parentheses; (2) *, **, and *** are significant at the level of 10%, 5%, and 1%, respectively.

**Table 5 ijerph-19-16842-t005:** Robustness test results.

The Explained Variable GS		
	(1)	(2)
INDS	0.503 ***	
	(4.513)	
TL		−0.258 ***
		(−2.799)
BS	0.035 ***	0.030 ***
	(3.762)	(2.722)
Popu	0.295 ***	0.196 ***
	(4.737)	(2.858)
Fian	0.022 *	0.016
	(1.908)	(0.940)
Per_gdp	0.162 **	0.16
	(2.392)	(1.324)
Park_Invest	0.042 ***	0.038 ***
	(5.862)	(5.309)
_cons	10.057 ***	13.274 ***
	(7.940)	(12.686)
Individual effect	control	control
Year effect	control	control
R—Square	0.916	0.937
N	2750	2750

Note: (1) t value in parentheses; (2) *, **, and *** are significant at the level of 10%, 5% and 1%, respectively.

**Table 6 ijerph-19-16842-t006:** Heterogeneity analysis results of east, central and west China.

	(1) In the East	(2) In the Middle	(3) In the West	(4) In the East	(5) In the Middle	(6) In the West
INDS	0.424 *	0.892 ***	0.038			
	(1.756)	(7.151)	(0.336)			
TL				−0.074	−0.462 **	−0.107
				(−0.431)	(−2.010)	(−0.990)
BS	0.028	0.033	0.044 ***	0.028	0.029	0.043 ***
	(1.396)	(1.611)	(2.942)	(1.485)	(0.927)	(2.831)
Popu	0.350 ***	0.312 ***	0.243 **	0.239 ***	0.053	0.237 **
	(3.793)	(2.701)	(2.238)	(3.507)	(0.257)	(2.171)
Fian	0.018	0.025	−0.022	0.018	0.022	−0.023
	(1.120)	(1.066)	(−1.089)	(1.296)	(0.693)	(−1.157)
Per_gdp	0.178 *	0.235 ***	0.070	0.112	−0.069	0.065
	(1.682)	(3.103)	(1.349)	(1.184)	(−0.413)	(1.353)
Park_Invest	0.044 **	0.052 ***	0.023	0.043 **	0.053 ***	0.023
	(2.631)	(3.200)	(1.489)	(2.589)	(3.218)	(1.503)
_cons	9.550 ***	8.514 ***	11.766 ***	12.193 ***	16.163 ***	12.048 ***
	(4.424)	(4.363)	(7.784)	(8.243)	(5.129)	(8.145)
Individual effect	control	control	control	control	control	control
Year effect	control	control	control	control	control	control
R—Square	0.607	0.883	0.517	0.641	0.916	0.539
N	968	1034	891	968	1034	891

Note: (1) t value in parentheses; (2) *, **, and *** are significant at the level of 10%, 5% and 1%, respectively.

## Data Availability

No new data were created or analyzed in this study. Data sharing is not applicable to this article.

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
