# Peer review of "Empirical Study on the Influence of Urban Environmental Industrial Structure Optimization on Ecological Landscape Greening Construction"

_ijerph, 2022, doi:10.3390/ijerph192416842_

Round 1

Reviewer 1 Report

1) From line 37 to line 43, there are too many short sentences in this part, which are somewhat incoherent and can be expressed more continuously.

2) From line 65 to line 68, there are some problems in the expression of this part. The meaning is not smooth and needs to be adjusted properly

3) A "." should be added after the table title number, such as "Table 1. Variable definition table". 

4) The format of the tables is not uniform, and it is recommended that the contents of the tables be arranged uniformly in the middle

5) Some simple text introduction is required at the top of Table 2.

6) This article has rich data analysis, which is very good, but there are some simple expression problems in the result analysis. The meaning is not very clear, so it can be said more clearly.

7) From line 362 to line 363, please check whether the expression of this sentence is correct.

8) On line 366, this position should not be marked with a citation.

9) Line 371, the time does not correspond to the previous text, please confirm.

Reviewer 2 Report

Review:

Manuscript ID: ijerph-1976093

Title: Empirical study on the influence of urban environmental industrial structure optimization on ecological landscape greening construction

Abstract:

Line 16 - ecological environment? maybe "natural environment"

 1. Introduction:

Line 36: "... has become more and more obvious.[1].."  - the end of the sentence should be each time after the reference to the literature [1]., so please be careful with that, as there are plenty of similar mistakes in the whole manuscript.

In the text there is often a lack of spaces between separate words.

I suggest moving the part of the text (lines 71-136) in to the Introduction.

Lines 357-359 - be more specific - which analyses?

Some sentences are also unclear. All the lines of the manuscript should be check, in order to remove the shortcomings that are found in the text.

In the manuscript there is a lack of proper discussion - maybe some part of  "4. Conclusions and Suggestions", should be rewritten for that purpose.

Reviewer 3 Report

<Comments to the authors>

Statistical assessments are important for the development of a sustainable society in order to determine environmental, economic, and urban development policies. It is interesting to investigate the relationship between environmental industrial structure adjustments ("advanced" vs. "rationalization") and greening. We hope you will consider our comments on this study, which are presented below.

<Specific comments>

<Major>

This study provides a statistical test of Hypothesis 1: Optimization of urban environmental industry structure has a positive impact on the construction of ecological landscape greening. Using panel data from China, a two-way fixed effect model of time individual was used to evaluate this hypothesis. The specific statistics used in this analysis would be easier to understand and more reliable. By making the analytical methods available on internet, I think it will facilitate the use of this study's approach in policy making toward a sustainable society.5.2. Although the proposal is abstract, the specifics of "advanced" and "rationalization" and "greening" to each region with different levels of economic development identified in this study for "advanced" and "rationalization" and "greening" are I think the proposal would be a meaningful discussion.

<Minor>

Detailed comments are as follows.

Table 6: Why is (5) a particularly low R-Square result?

4. the item numbers in the Conclusions and Suggestions are incorrect.

Round 2

Reviewer 1 Report

The authors have addressed my comments properly.